# Detwinning Mechanism for Nanotwinned Cubic Boron Nitride with Unprecedented Strength: A First-Principles Study

**DOI:** 10.3390/nano9081117

**Published:** 2019-08-03

**Authors:** Bo Yang, Xianghe Peng, Sha Sun, Cheng Huang, Deqiang Yin, Xiang Chen, Tao Fu

**Affiliations:** 1Department of Engineering Mechanics, Chongqing University, Chongqing 400044, China; 2State Key Laboratory of Coal Mining Disaster Dynamics and Control, Chongqing University, Chongqing 400044, China; 3Advanced Manufacturing Engineering, Chongqing University of Posts and Telecommunications, Chongqing 400065, China

**Keywords:** nanotwin, detwinning, extreme hardness, excellent stability

## Abstract

Synthesized nanotwinned cubic boron nitride (nt-cBN) and nanotwinned diamond (nt-diamond) exhibit extremely high hardness and excellent stability, in which nanotwinned structure plays a crucial role. Here we reveal by first-principles calculations a strengthening mechanism of detwinning, which is induced by partial slip on a glide-set plane. We found that continuous partial slip in the nanotwinned structure under large shear strain can effectively delay the structural graphitization and promote the phase transition from twin structure to cubic structure, which helps to increase the maximum strain range and peak stress. Moreover, ab initio molecular dynamics simulation reveals a stabilization mechanism for nanotwin. These results can help us to understand the unprecedented strength and stability arising from the twin boundaries.

## 1. Introduction

Diamond and zinc blende structured materials are the important members in the family of superhard materials, among which diamond and cubic boron nitride (cBN) are the most prominent representatives. These strong covalent bond solids are indispensable to fundamental scientific research and technological applications in many fields [1,2,3,4,5,6,7,8]. With the development of synthesis technology at nanoscale, more and more attention has been paid to nanocrystalline (NC) superhard materials as well as their outstanding mechanical properties [9,10,11,12,13]. It has been found that grain size plays a decisive role in the strength of materials [14,15,16]. For example, the strength of NC Cu increases with the decrease of grain size, and reaches the maximum at the critical size (*d* = 19.3 nm), followed by softening with the further decrease of grain size [17]. Similar phenomena have also been observed in covalently bonded materials, for example, at room temperature the Knoop hardness of an NC diamond with grain size of 10–20 nm is about 130 GPa, much higher than that of single crystal diamond, which is about 70–90 GPa [18,19], and the strength of NC cBN increases with the decrease of grain size *d* [20,21], following the famous Hall-Petch relationship [15,22].

Recent studies [23] showed that nanotwinned cBN (nt-cBN) and nanotwinned diamond (nt-diamond) exhibit extremely high mechanical and thermal properties compared with their single crystal counterparts. The Vickers hardness of nt-cBN with an average twin thickness of *λ* = 3.8 nm reaches 95–108 GPa, which even exceeds that of a single crystal diamond (90 GPa) [23]. This high hardness was attributed to the existence of high-density nanotwins [24]. On the other hand, the onset oxidation temperature of nt-cBN is also about 200 °C higher than that of single cBN crystal. The corresponding studies [23,25] challenge the general understanding of the properties of materials at nanoscale. It has been shown that, in nanotwinned metals [26,27,28], the appropriate thickness and distribution of twins play significant roles in the improvements of the physical and mechanical properties of metals, including strength, hardness, and thermal stability. However, it has been shown in many studies that the strengthening/weakening mechanisms between nt-metals and nt-cBN are obviously different [24,29,30]. At present, the role of nanotwins in the enhancement of the mechanical properties of strong covalent bond solids remains controversial [31,32] and needs to be clarified.

In this work, the mechanisms for the enhancement of mechanical properties and stability of nt-cBN under continuous shear strain are investigated using first-principles calculations. The paper is organized as follows: the calculation details are briefly introduced in the following section, followed by the presentation of the results calculated and the corresponding discussions, and some conclusions are drawn and shown in the last section.

## 2. Methods

In this work, the stability and mechanical properties of materials are described by the stress-strain relationships, twin boundary energies (TBEs) and ab initio molecular dynamics (MD) simulations, respectively. All simulations are performed using the Vienna ab initio simulation package (VASP) code based on the density functional theory (DFT) with the generalized gradient approximation of Perdew-Burke-Ernzerh of version [33] for the exchange-correlation energy and a plane-wave basis set [34]. The projector augmented wave (PAW) method describes the electron-ion interaction [35], in which 2*s*^2^2*p*^1^, 2*s*^2^2*p*^2^, and 2*s*^2^2*p*^3^ are the valence electrons for B, C, and N atoms, respectively. An energy cutoff of 500 eV and Monkhorst-Pack *k*-point grids [36] of 5 × 7 × 3 are used for the calculation of the differences in the total energy and stress response of perfect crystal and twin structure. The convergence criterion is that the total energy of the system and the force on each of the atoms are less than 1 × 10^−4^ eV and 0.001 eV/Å, respectively.

We utilize standard ab initio MD simulations as implemented in the VASP code to observe insights into the equilibrium structure of diamond and cBN at a given temperature. The setting accuracy of calculation parameters is consistent with that of the basic properties of the above DFT energy calculation. Moreover, in the intermediate period, a micro-canonical ensemble (NVE) is simulated. A supercell containing 96 atoms is employed to avoid artifacts associated with constraints imposed by finite-sized unit cells [37]. In addition, the parameters related to the simulation time are set to 3 femtoseconds (fs) per time step and the maximum ionic step is 1500.

To obtain the failure shear stress, we apply a shear strain component on a crystal cell along the prescribed orientation, while relaxing the other five strain components, and deform the periodic model until it fails [7,38,39,40]. To determine the lowest energy path during deformation, we calculate the generalized stacking fault energy (GSFE) surface, also called *γ*-surface, which describes the energy variation between that of the perfect crystal and that of the half-crystal shift *d* on the crystallographic plane prescribed [41]. For example, to obtain the *γ*-surface of the (111) plane of cBN, we divide the plane of cBN into 525 grids along the $\left[11\bar{2}\right]$ and $\left[1\bar{1}0\right]$ directions, calculate the difference between the energy of the perfect crystal and that with the origin of the upper-half shifted by the distance to each grid.

## 3. Results and Discussions

### 3.1. Excellent Stability of Nanotwinned Structure

Figure 1 shows schematically the atomic arrangement of perfect crystal and twin of cBN. In order to measure the stability of the twin structure, we calculate the twin boundary energy (TBE), which represents the energy variation between the perfect crystal and twin structure of per unit area. The smaller the TBE, the better the stability of the twin structure [38]. The TBEs of cBN and diamond are calculated and listed in Table 1, where one can see that the TBEs of cBN and diamond are 81.72 and 101.69 mJ/m^2^, respectively. Moreover, the total energy of crystalline cBN and that of twined cBN are basically the same, so are those of diamond. The above information indicates that twin structure should be as stable as its single crystal for cBN and diamond. However, the extremely high GSFE may limit the formation of the twin structure in cBN. The recent report [23,25] of nt-cBN converted from cBN nano-onions with high-density defects under high temperatures and pressures solves this synthetic problem.

To reveal the contribution of the nanotwinned structure to the unusual thermal stability of the strong covalent bond solid, ab initio MD simulations were carried out. We have qualitatively compared the thermal stability of diamond and nt-diamond at T = 1500 K. Results of our ab initio MD simulations are presented as structural snapshots of diamond and nt-diamond at T = 1500 K in Figure 2a,b. In addition, the Appendix A respectively describe ab initio molecular dynamics simulations of diamond and nt-diamond at T = 1500 K in the front view. We adopted a covalent bond cutoff radius of 1.5 Å for diamond and nt-diamond. As shown in Figure 2a,b, some C–C bonds of the diamond are broken while the C–C bonds of nt-diamond are basically intact. It is obvious that the structural change of nt-diamond is less than that of the diamond, indicating the twin structure does contribute to the stability of the diamond, which verifies the conclusion of the experiment. It should be noted that structural change refers to the difference between a new structure and the original structure. In parallel, the twin structure does not contribute significantly to the thermal stability of cBN, which may be related to the superior thermal stability of cBN itself [43,44]. Figure 2c,d, and Appendix A give more information.

It has been reported in the previous studies [45,46] that the (111) cleavage plane clearly dominates the fracture of cBN or diamond. The *γ*-surfaces in cBN for glide-set and shuffle-set planes are calculated for the determination of the lowest energy path during deformation, as shown in Figure 3a,b, respectively. It can be seen that the lowest energy path of $\frac{a_{0}}{6}[11\bar{2}]$ ($a_{0}=3.625\;\overset{o}{A}$, representing the lattice constant) for glide-set and shuffle-set planes, where the peak of the path is defined as the energy barrier (*γ_U_*). Then, the GSFE curves for glide-set and shuffle-set planes in (111)<112> slip system are extracted and shown in Figure 4a, where the *γ_Ug_* on the glide-set plane is 3.72 J/m^2^, which is smaller than that on the shuffle-set plane (6.85 J/m^2^), indicating that the slip of cBN should be dominated by the partial slip on the glide-set plane, which coincides with the conclusion that the stacking fault has been observed experimentally [47], as shown in Figure 3c–f. In order to understand more clearly the stacking faults in the experiment, the atomic configurations of several key structural points have been revealed. As shown in Figure 4b, the stacking sequence of the (111) planes of cBN can be expressed as AαBβCγAαBβCγ…. The upper part of the model is rigidly displaced along the $\left[11\bar{2}\right]$ direction on the glide-set plane. Figure 4c shows the atomic configuration when normalized displacement is equal to $\frac{a_{0}}{12}[11\bar{2}]$, which is unstable, corresponding to the unstable GSFE of *γ*_Ug_. When normalized displacement is equal to $\frac{a_{0}}{6}[11\bar{2}]$, as shown in Figure 4d, the stacking sequences becomes AαBβCαBβCγAα…, indicating the occurrence of a stacking fault. At this moment, the GSFE is at the valley bottom (*γ*_Ig_), which implies a metastable state. Therefore, the existence of a local nanotwined structure is also demonstrated by the principle of minimum energy.

### 3.2. Detwinning Mechanism of nt-cBN

Figure 5 shows the shear stress-strain (*σ*-*ε*) and energy-strain (*E*-*ε*) curves of nt-cBN during shear straining along the (111)$\left[11\bar{2}\right]$ direction, where both the curves exhibit a distinct zigzag feature. This zigzag feature is significantly different from cBN, as reported previously [48], fails directly during the shear straining due to graphitization. It can be seen in Figure 5 that *σ* and *E* increase quickly as the applied shear strain increases from *ε*_0_
*=* 0 to *ε*_1_
*=* 0.23 when the stress reaches 62.1 GPa, which is slightly smaller than the stress (67.2 GPa) before the graphitization of cBN subjected to shear deformation along the easy shear direction. When the strain increases from *ε*_1_
*=* 0.23 to *ε*_2_
*=* 0.24, the stress and total energy fall sharply. In order to clarify the cause of the zigzag manner, we extract the atomic configurations at some key points, as shown in Figure 6. It can be found that the root cause of this phenomenon is atomic reconfiguration induced by partial slip on the glide-set plane, in which the old chemical bond B_1_-N_1_-B_2_ is broken, and a new chemical bond B_2_-N_1_-B_3_ is formed, as shown in Figure 6a,b. 

The formation of the new metastable structures (Figure 6b) can further resist deformation induced by the further shear strain. From *ε*_2_
*=* 0.24 to *ε*_3_
*=* 0.43, *σ* increases from the valley of 3.7 GPa to the peak of 65.8 GPa. Then, the same atomic reconfiguration occurs in the layer C′, resulting in the break of the chemical bond B_7_-N_3_-B_8_, and the formation of chemical bond B_8_-N_3_-B_9_, as shown in Figure 6c,d. This process will be repeated during the shear straining until all the atomic layers in the easy shear direction are transformed into that in the hard shear direction, and, correspondingly, the sequence of the layers would change from ...ABC|C’B’A’... [Figure 6a] to ABCABC... [Figure 6e] when the detwinning of nt-cBN by partial slip is finished. The further increase of strain would induce the deformation along the hard shear direction of the cBN, as shown in Figure 6e, and at *ε*_5_
*=* 0.59 the ultimate stress reaches 87.2 GPa, which is almost equal to the graphitization stress (87.4 GPa) of cBN along the hard shear direction, as shown in Figure 5. Lattice instability occurs as ε ≥ ε_6_ = 0.60, leading to graphite-like layered structures, as shown in Figure 6f. 

The above process describes accurately the one-to-one correspondence between *σ* (or *E*) and atomic reconfiguration. In general, such kind of detwinning mechanism can lead to an unprecedented increase in intragranular deformation resistance for nt-cBN. More detailed information on the atomic reconfiguration during the deformation of cBN and nt-cBN are shown in Appendix A. Appendix A show the structural changes along the easy and hard shear directions of cBN subjected to shear deformations, respectively, where one can see that cBN fails directly by graphitization, which is consistent with the previous report [47]. By contrast, the detwinning mechanism of nt-cBN by partial slip on the glide-set plane is shown in Appendix A, where the continuous partial slip can effectively delay the structural graphitization, contributing to the remarkable increases of the maximum strain and intragranular deformation resistance.

To verify the fascinating phase transition from nt-cBN to cBN, we unload the stress from *ε*_4_ = 0.44 by decreasing the strain until *σ* = 0, as shown in Figure 5. After unloading, the lattice vectors of the cell are along the $\left[\bar{1}\bar{1}2\right]$, $\left[1\bar{1}0\right]$ and $\left[11\bar{2}\right]$ directions, which are exactly identical with those of the perfect crystal cBN cell, indicating the nt-cBN has been completely detwinned. Figure 7 shows the comparison between the atomic structure of relaxed cBN at zero stress and that of detwinned nt-cBN after unloading, where there are two findings. First, the positions of the atoms in the two structures overlap completely, second, the angle between lattice vectors $\left[\bar{1}\bar{1}2\right]$ and $\left[11\bar{2}\right]$ is approximately 70.55°, which is consistent with the experimental observations [47], as shown in Figure 3f.

## 4. Summary

We reported a detwinning mechanism for nt-cBN, which may result in extremely high hardness, which can describe accurately the one-to-one correspondence between the stress or the total energy and the atomic reconfiguration of nt-cBN subjected to shear straining. The atomic layers in the easy shear direction can be shifted into the hard shear direction by partial slip, which also leads to the decrease of the stress and total energy of the system and formation a new stable structure. This characteristic should be conducive to the stability of the structure undergoing lager shear strain. Moreover, the twin boundary energies of nt-diamond and nt-cBN are very low, which indicates that the twin structure is as stable as its single crystal counterparts. The excellent thermal stability of nt-diamond and nt-cBN has also been systematically studied by ab initio molecular dynamics (MD) simulations and a stabilization mechanism for nanotwin was revealed. These results can not only account for the unprecedented hardness and excellent stability of nanotwinned structure but offer novel insights into the deformation-induced structural transformation as well. It can also extend our understanding of the deformation mechanism of nanostructured strong covalent materials, which would be useful in guiding the design of ultrahard materials.

## Figures and Tables

**Figure 1 nanomaterials-09-01117-f001:**
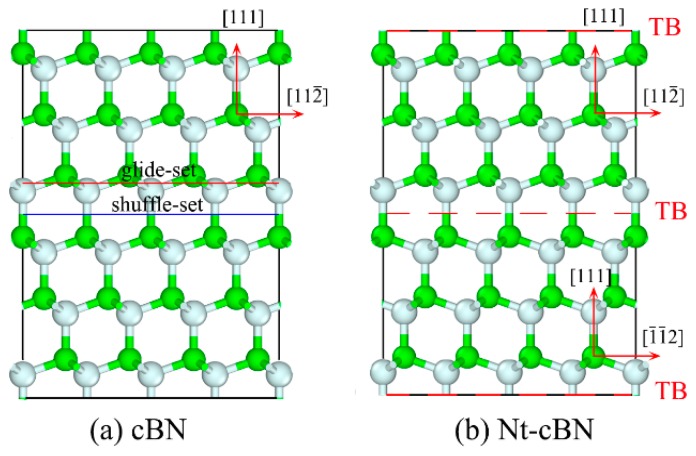
Atomic arrangement of cubic boron nitride (cBN) and twin cBN: (**a**) cBN, with two non-equivalent (111) planes, corresponding respectively to narrowly spaced atomic layer (glide-set) indicated by red line, and widely spaced atomic layer (shuffle-set) by blue line [42]. (**b**) nanotwinned cubic boron nitride (nt-cBN) with red lines as twin boundary (TBs).

**Figure 2 nanomaterials-09-01117-f002:**
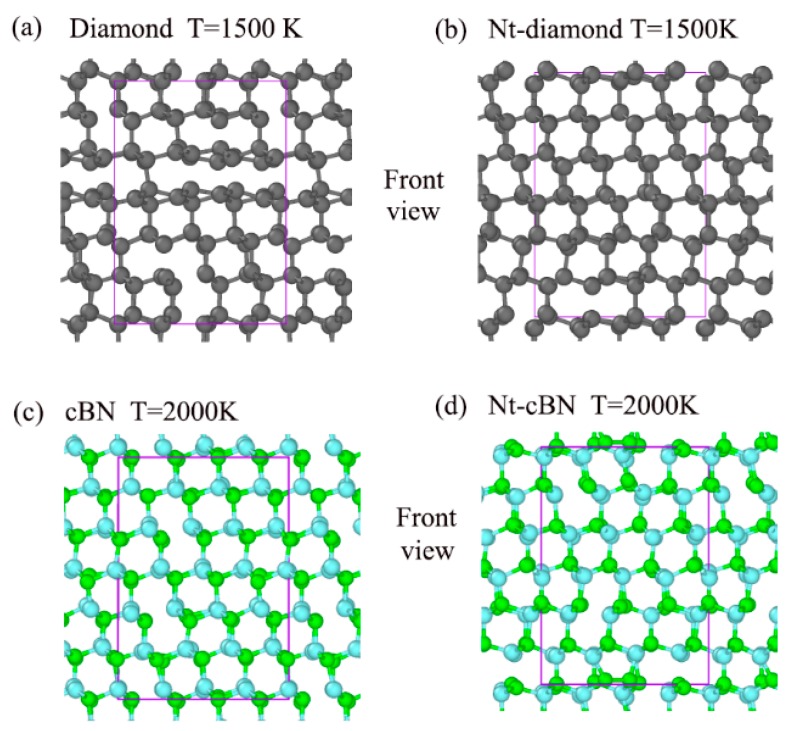
(**a**) and (**b**) are snapshots of ab initio molecular dynamics simulations depicting structural changes in diamond and nt-diamond at T = 1500 K. Corresponding snapshots of cBN and nt-cBN at T = 2000 K are shown in (**c**) and (**d**). The solid purple line represents unit cell.

**Figure 3 nanomaterials-09-01117-f003:**
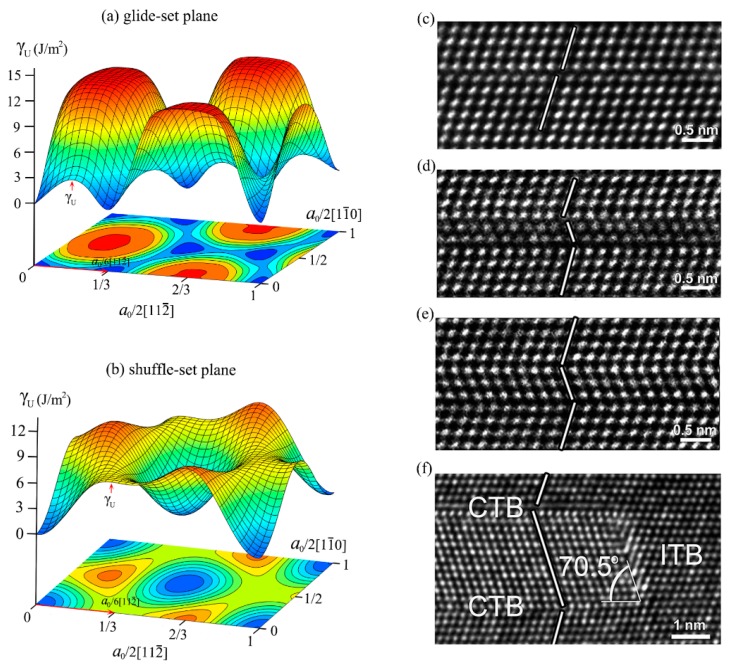
*γ*-surface and stacking fault region of cBN. (**a**) *γ*-surface of glide-set plane and (**b**) *γ*-surface of shuffle-set plane, with *γ_U_* denoting energy barrier, red arrows indicating lowest energy path, respectively, (**c**–**f**) stacking fault region in cBN observed in experiment. Reproduced with permission from [47], copyright AIP Publishing, 2016.

**Figure 4 nanomaterials-09-01117-f004:**
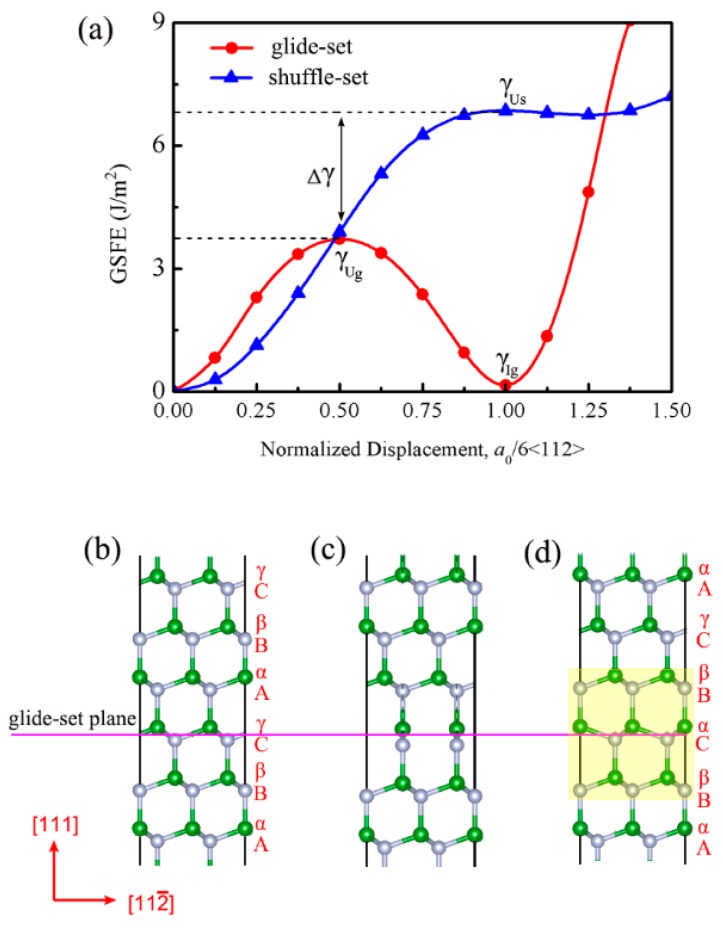
(**a**) Calculated GSFE curves for cBN in (111)<112> slip system, where γ_Ug_ and γ_Us_ denote energy barrier on glide-set and shuffle-set planes, respectively. Atomic configurations corresponding to three key structures in (111)<112> slip system: (**b**) initial configuration, (**c**) unstable configuration, and (**d**) metastable configuration, where light-yellow regions correspond to the stacking fault region.

**Figure 5 nanomaterials-09-01117-f005:**
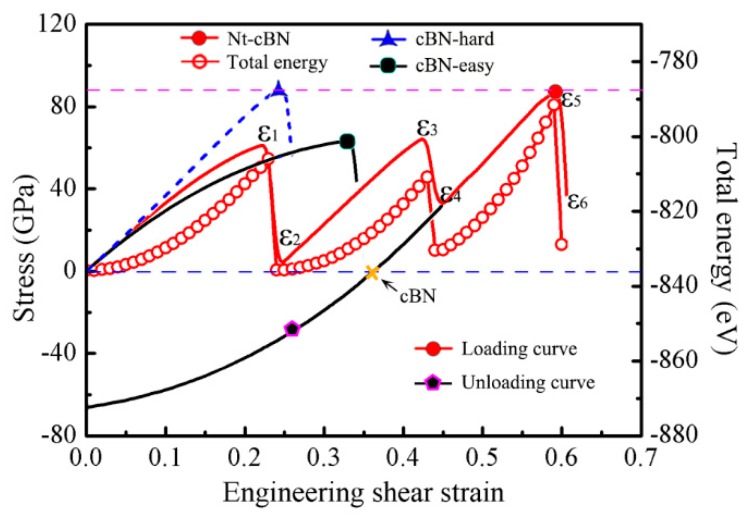
Calculated *σ*-*ε* and *E*-*ε* curves of nt-cBN subjected to shear straining and unloading, *σ-ε* curves of cBN sheared along (111)$\left[11\bar{2}\right]$ easy shear direction and (111)[$\bar{1}\bar{1}2$] hard shear direction also provided for comparison.

**Figure 6 nanomaterials-09-01117-f006:**
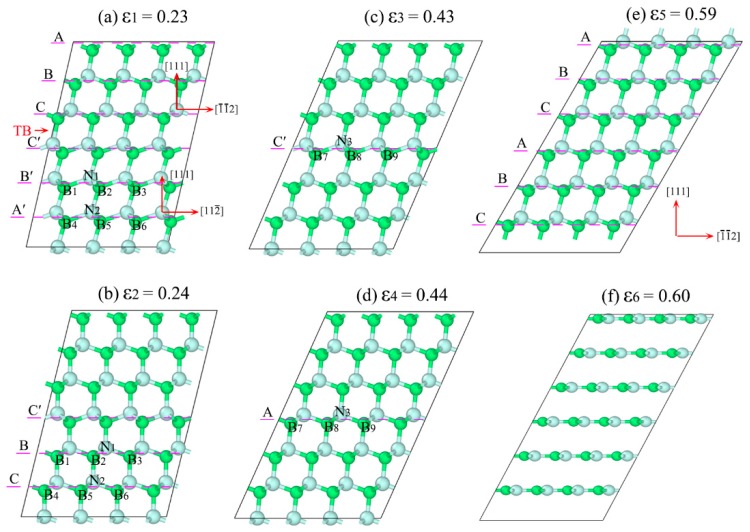
(**a**) to (**f**) are typical atomic configurations of nt-cBN sheared along (111)$\left[11\bar{2}\right]$ easy shear direction during 0.23 ≤ ε ≤ 0.60, respectively, with upper and lower halves corresponding to easy- and hard-shear directions in (**a**).

**Figure 7 nanomaterials-09-01117-f007:**
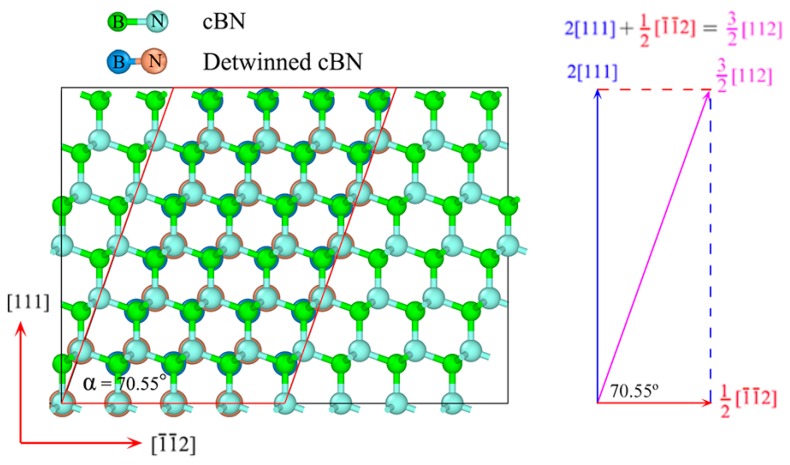
Comparison between atomic structure of relaxed cBN at zero stress and that of detwinned cBN after unloading.

**Table 1 nanomaterials-09-01117-t001:** Calculated total energies and TBEs of cBN and diamond.

	Total Energy (ev)		TBE (mJ/m^2^)
	Perfect Crystal	Twin	
cBN	−836.29	−835.83	81.72
Diamond	−872.62	−872.06	101.69

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
