# Peer review of "Detwinning Mechanism for Nanotwinned Cubic Boron Nitride with Unprecedented Strength: A First-Principles Study"

_nanomaterials, 2019, doi:10.3390/nano9081117_

Reviewer 1 Report

The paper brings interesting new results on detwinning mechanisms in nanotwinned cubic boron nitride and elucidates the role of twins under large shear strains. It is written concisely and comprehensively and the results are sufficiently novel and of broad interest to warrant the publication in Nanomaterials.
Some minor items which should be corrected before publication:
1. p. 2, lines 51-53:
The authors write:
In this work, the mechanisms for the enhancement of the mechanical property and stability of nt-cBN under continuous shear strain are investigated using first principles calculations. The content is arranged as follows:
A better formulation:
In this work, the mechanisms for the enhancement of mechanical properties and stability of nt-cBN under continuous shear strain are investigated using first principles calculations. The paper is organized as follows:
2. p. 2, line 57:
The authors write:
In this work, the mechanical and stable properties of materials ...
A better formulation:
In this work, stability and mechanical properties of materials ...
3. p. 2, line 61: not "Perdew-Burke- Ernzerh of", but "Perdew-Burke-Ernzerhof"
4. p. 2, line 64: not "Monkhorst-pack", but "Monkhorst-Pack"
5. p. 2, line 66: not "the total energy", but "differences in the total energy"
6. p. 8, Fig. 7: The description in the right-hand side of Fig. 7 is hardly legible - I recommend somewhat larger numerals.
7. pp. 9-11: Some letters in the references should be capitalized, e.g.
p. 9, line 248:  Physical Review Letters
p. 10, line 321:  not "mc systems (m = ti, zr, hf)", but "MC systems (M = Ti, Zr, Hf)"  etc.
After these corrections, the paper may be published in Nanomaterials.

Author Response

Response to the comments from Reviewer #1

The paper brings interesting new results on detwinning mechanisms in nanotwinned cubic boron nitride and elucidates the role of twins under large shear strains. It is written concisely and comprehensively and the results are sufficiently novel and of broad interest to warrant the publication in Nanomaterials.
Some minor items which should be corrected before publication:

We thank the reviewer for the positive evaluation and the recommendation of the possible publication of our manuscript. The following is our response to each comment.

(1) p. 2, lines 51-53:
The authors write:
In this work, the mechanisms for the enhancement of the mechanical property and stability of
nt-cBN under continuous shear strain are investigated using first principles calculations. The content is arranged as follows:
A better formulation:
In this work, the mechanisms for the enhancement of mechanical properties and stability of
nt-cBN under continuous shear strain are investigated using first principles calculations. The paper is organized as follows:

(2) p. 2, line 57:
The authors write
:
In this work, the mechanical and stable properties of materials ...
A better formulation:
In this work, stability and mechanical properties of materials ...

(3) p. 2, line 61: not "Perdew-Burke- Ernzerh of", but "Perdew-Burke-Ernzerhof"

(4) p. 2, line 64: not "Monkhorst-pack", but "Monkhorst-Pack"

(5) p. 2, line 66: not "the total energy", but "differences in the total energy"

(6) p. 8, Fig. 7: The description in the right-hand side of Fig. 7 is hardly legible - I recommend somewhat larger numerals.

(7) p. 9-11: Some letters in the references should be capitalized, e.g.
     p. 9, line 248: Physical Review Letters
     p. 10, line 321: not "mc systems (m = ti, zr, hf)", but "MC systems (M = Ti, Zr, Hf)" etc.
    After these corrections, the paper may be published in Nanomaterials.

We thank the reviewer for the valuable comments and suggestions. We apologize for our carelessness and the mistakes. We have carefully revised the manuscript in accordance with the comments from the reviewer, and the changes made are expressed in red in the revised manuscript

Reviewer 2 Report

Peng, Fu et al. report on ab initio calculations used to rationalize the properties and detwinning mechanism for synthesized nanotwinned cubic BN. This is an interesting study, worth publishing, however, some revision points need to be addressed.
(1) Page 2, Methods: please specify which "standard" ab initio MD simulation was used. Please mention ensemble (NVT), the precision used (normal, low...), as well as any change in parameters compared to the DFT energy computation. This can help to ensure reproducibility of calculations for people using other codes.
(2) Page 2, line 90: please quote literature or present the supporting calculations.
(3) Page 4, line 110 ("structural change..."): please quantify this statement.
(4) Page 1, line 37: please quote reference for Hall-Petch relationship.
Page 1, line 41: please quote the hardness values for diamond as well.

Author Response

Response to the comments from Reviewer #2

Peng, Fu et al. report on ab initio calculations used to rationalize the properties and detwinning mechanism for synthesized nanotwinned cubic BN. This is an interesting study, worth publishing, however, some revision points need to be addressed.

We thank the reviewer for the positive evaluation and the recommendation of the possible publication of out manuscript. The following is our response to each comment.

(1) Page 2, Methods: please specify which "standard" ab initio MD simulation was used. Please mention ensemble (NVT), the precision used (normal, low...), as well as any change in parameters compared to the DFT energy computation. This can help to ensure reproducibility of calculations for people using other codes.

We thank the reviewer for the valuable suggestion. As the reviewer said, the setting of calculation parameters is very critical. In our ab initio MD simulation, the setting accuracy of calculation parameters is consistent with that of the basic properties of the DFT energy computation. Moreover, in the intermediate period a micro–canonical ensemble (NVE) is simulated.  We have added above discussions  to the revised manuscript.

(2) Page 2, line 90: please quote literature or present the supporting calculations.

     We thank reviewer very much for the kind reminder. We have added references to the corresponding parts of the reserved manuscript.

(3)  Page 4, line 110 ("structural change..."): please quantify this statement.

      We thank the reviewer for the kind suggestion. The structural change refers to the difference between a new structure and the original structure. We adopted a covalent bond cutoff radius of 1.5 Å for diamond and nt-diamond. As shown in Figs. R1 (a) and (b), some C-C bonds of diamond are broken while the C-C bonds of nt-diamond are basically intact. It is obvious that the structural change of nt-diamond is less than that of diamond, indicating the twin structure does contribute to the stability of the diamond, which verifies the conclusion of the experiment. We have added this definition to the revised manuscript.

Fig. R1 (a) and (b) are snapshots of ab initio molecular dynamics simulations depicting structural changes in diamond and nt-diamond at T= 1500 K. Corresponding snapshots of cBN and nt-cBN at T = 2000 K are shown in (c) and (d). The solid purple line represents unit cell.

(4) Page 1, line 37: please quote reference for Hall-Petch relationship.
Page 1, line 41: please quote the hardness values for diamond as well.

We thank reviewer very much for the kind reminder. We have added the references and the hardness values of diamond to the corresponding parts of the reserved manuscript.
